# Urinary Tract Infections Impair Adult Hippocampal Neurogenesis

**DOI:** 10.3390/biology11060891

**Published:** 2022-06-09

**Authors:** Batoul Darwish, Farah Chamaa, Bassel Awada, Nada Lawand, Nayef E. Saadé, Antoine G. Abou Fayad, Wassim Abou-Kheir

**Affiliations:** 1Department of Anatomy, Cell Biology and Physiological Sciences, Faculty of Medicine, American University of Beirut, Beirut P.O. Box 11-0236, Lebanon; bmd09@mail.aub.edu (B.D.); farah.chamaa@kaust.edu.sa (F.C.); nl08@aub.edu.lb (N.L.); nesaade@aub.edu.lb (N.E.S.); 2Biological and Environmental Science and Engineering Division, King Abdullah University of Science and Technology, Thuwal 23955, Saudi Arabia; 3Department of Experimental Pathology, Immunology and Microbiology, Faculty of Medicine, American University of Beirut, Beirut P.O. Box 11-0236, Lebanon; baa60@mail.aub.edu (B.A.); aa328@aub.edu.lb (A.G.A.F.); 4Center for Infectious Diseases Research, American University of Beirut, Beirut P.O. Box 11-0236, Lebanon; 5Department of Neurology, Faculty of Medicine, American University of Beirut, Beirut P.O. Box 11-0236, Lebanon

**Keywords:** inflammation, dentate gyrus, neural stem cells, heat hyperalgesia, BDNF, NGF, cognitive behavior

## Abstract

**Simple Summary:**

Urinary tract infections are associated with features of cognitive decline and memory deficits, where the underlying correlation or mechanism is still not clear. In this study, we investigate the effect of urinary tract infections on cognitive functions in rodents and whether it is associated with adult hippocampal neurogenesis, a process that is detrimental for memory formation. We have shown that urinary tract infection affects the time spent exploring a novel arm in the Y-maze test. This was accompanied with a decrease in the proliferation of neural stem cells at an early time point post infection and a persistent decrease in neurogenesis at a later time point (34 days). We also detected decreased levels of neurotrophic factors important for neurogenesis and an elevated expression of interleukin 1β in the hippocampus. Treatment with either anti-inflammatory drugs or anti-biotics does not recover proliferation of neural stem cells. Here, we present hippocampal neurogenesis as a possible contributor to cognitive changes associated with urinary tract infections. Given the significant increase in urinary tract infection occurrence, it is important to address some of the detrimental effects that such an infection can have at the level of the brain.

**Abstract:**

Previous studies have suggested a link between urinary tract infections (UTIs) and cognitive impairment. One possible contributing factor for UTI-induced cognitive changes that has not yet been investigated is a potential alteration in hippocampal neurogenesis. In this study, we aim to investigate the effect of UTI on brain plasticity by specifically examining alterations in neurogenesis. Adult male Sprague Dawley rats received an intra-urethral injection of an *Escherichia coli* (*E. coli)* clinical isolate (10^8^ CFU/mL). We found that rats with a UTI (CFU/mL ≥ 10^5^) had reduced proliferation of neural stem cells (NSCs) at an early time point post infection (day 4) and neurogenesis at a later time point (day 34). This was associated with the decreased expression in mRNA of BDNF, NGF, and FGF2, and elevated expression of IL-1β in the hippocampus at 6 h post infection, but with no changes in optical intensity of the microglia and astrocytes. In addition, infected rats spent less time exploring a novel arm in the Y-maze test. Treatment with an anti-inflammatory drug did not revert the effect on NSCs, while treatment with antibiotics further decreased the basal level of their proliferation. This study presents novel findings on the impact of urinary tract infections on hippocampal neurogenesis that could be correlated with cognitive impairment.

## 1. Introduction

Adult hippocampal neurogenesis is the process of the formation and integration of new neurons from the neural stem cells (NSCs) housed within the neurogenic niche of the dentate gyrus (DG) of the hippocampus [1]. This process is instrumental for learning, memory encoding, and cognitive function [2]. One of the most intriguing characteristics of hippocampal neurogenesis is that it is highly prone to changes in response to a variety of intrinsic and extrinsic factors [3]. Among these factors, inflammation, whether general or localized in the brain (neuroinflammation), is a well-known strong suppressor of adult hippocampal neurogenesis [4,5,6]. The brain has been previously assumed as an immune-privileged organ. However, the reciprocal interactions between the brain and the immune system are becoming more evident and apparent with more experimental evidence unraveling their interplay [7,8]. Inflammation in the periphery has been previously associated with decreased neurogenesis and cognitive changes [9,10,11,12,13]. As an illustration, recent clinical evidence tends to attribute the reported memory deterioration experienced during and after COVID-19 viral infection to ongoing inflammation [14,15,16]. Inflammation is also strongly elicited in response to urinary tract infections (UTIs), which have been often associated, in elderly patients, with delirium and memory deterioration that usually subside after treatment.

Recent studies have shown that UTIs are among the most commonly occurring bacterial infections associated with cognitive impairment [17,18]. Among elderly patients, delirium is considered a non-specific symptom of UTI and is the most common reason for suspecting UTIs in such an age group [19,20,21]. In addition, increasing evidence suggests that UTIs in elderly patients might increase the risk of developing dementia, or even aggravate pre-existing dementia [22,23]. However, such knowledge has been based on clinical observation without attempting to investigate the cellular and molecular mechanisms leading to such changes. So far, the exact mechanisms correlating the effects of UTI on cognitive decline are not yet fully explored [17,24]. Taking into account the role of neurogenesis in memory formation, we suspect that neurogenesis could be altered during UTI and might be considered as one of the factors behind these clinical symptoms. Thus, we aim to investigate the effect of UTIs on hippocampal neurogenesis and cognitive function in adult male rodents. For this purpose, we designed a rat model for UTI to investigate its effect on neurogenesis and behavioral tests for nociception and cognitive behavior.

## 2. Materials and Methods

### 2.1. Animals

Adult male Sprague Dawley rats, 3 months of age and weighing 450–550 g were employed. The rats were housed under standard colony conditions in a room maintained at a constant temperature (20–22 °C) on a 12 h light/dark cycle with standard rodent chow and water provided ad libitum. Animals were habituated to housing conditions a week prior to the experiment. All experimental procedures were approved by the Institutional Animal Care and Use Committee (IACUC approval number: 19-09-546) at the American University of Beirut (AUB).

### 2.2. Experimental Groups and Design

Figure 1 provides a summary of the different experimental groups and their time of treatment and sacrifice (Figure 1). Rats were sacrificed at 6 and 24 h (h) post infection to investigate the early inflammatory response, while others were sacrificed on day 4 or day 34 post infection to investigate the proliferation of DG neural stem cells and neurogenesis, respectively. Moreover, rats with a UTI and the sham group received treatment with either the non-steroidal anti-inflammatory drug (NSAID) Piroxicam or the antibiotic drug Fosfomycin and were sacrificed on day 4 post infection. All groups and sacrifice time points are listed in Table 1.

### 2.3. Urinary Tract Infection

An *Escherichia coli* (*E. coli*) 1176 clinical isolate obtained from the urine of a patient at the American University of Beirut Medical Center (AUBMC) was used in this study. The experimental procedure for inflicting urinary tract infection was performed as described previously by Zychlinsky Scharff et al., 2017, with the single adjustment of the injection volume from 50 µL in mice to 500 µL in rats. Briefly, a single colony of an overnight culture of *E. coli* 1176 on MacConkey agar was used to inoculate 10 mL of Luria Bertani (LB) broth. The resulting culture was incubated statically overnight at 37 °C. The bacterial pellet was resuspended in sterile phosphate buffered saline (PBS) in order to obtain a bacterial suspension with an optical density (OD_600_) of 0.5 McFarland, which is equivalent to 10^8^ colony forming units (CFU) per mL. Intraurethral injections were performed under isoflurane anesthesia and a compression was applied on the abdomen overlaying the bladder to allow any present urine to be released. The catheters used were the BD Instyle Autoguard shielded IV catheters (24 G, 0.7 mm external diameter, 14 mM long) (Figure 1B). The catheter should slide smoothly into the urethra, indicating correct placement, which was tested on rats by injecting trypan blue (Figure 1C). Sham rats received an intra-urethral injection of equal volume of 500 µL of a PBS vehicle. Urine was collected after 24 h post bacterial infection. Serial dilutions in LB broth were immediately performed to establish CFU counts from urine. CFU/mL assay was performed on MacConkey agars and plates were kept overnight in an incubator at 37 °C. Colonies were counted the next day from dilutions that had the number of colonies between 5 and 150 and CFU/mL, which was calculated as follows:(1)$${\mathrm{CFU}/\mathrm{mL}\;}_{\mathrm{Urine}}\;=\frac{N\;\times \;\mathrm{Dilution}\;\mathrm{Factor}}{\mathrm{Plated}\;\mathrm{volume}\;\left(\mathrm{mL}\right)\;}$$
where N is the number of counted bacterial colonies.

### 2.4. Treatment Regimens with the Antibiotics (Fosfomycin) and the NSAIDs (Piroxicam)

Rats were treated with Fosfomycin (200 mg/kg, i.p.) twice daily over 4 days starting from the time of intraurethral bacterial injection. The injections were separated by 8 h with the first given one hour post infection. This treatment regimen is similar to clinical settings and represents adequate dosage exposure for Sprague Dawley rats [25]. Fosfomycin is a broad-spectrum antibiotic used mainly for the treatment of uncomplicated lower urinary tract infections and is effective against multi-drug resistant (MDR) bacterial strains. We selected Fosfomycin since the clinical isolate *E. coli* 1176 used in this study was resistant to all antibiotics except Fosfomycin and Colistin. Colistin was not chosen due to its suspected nephrotoxicity and neurotoxicity [26,27].

Rats were treated with Piroxicam (10 mg/kg, i.p, Feldene©) over 4 days. On the day of infection, the 1st dose of Piroxicam was given 30 min before intra-urethral injection of *E. coli* and the second dose was given 5 h post infection; this was followed by only one injection over the next 3 days. Piroxicam dosage was based on previous evidence from our group showing that daily injections of 12 mg/kg, i.p. do not affect basal levels of neurogenesis [28]. Moreover, Piroxicam is an NSAID drug and a member of the oxicam group. It has a long plasma half-life of approximately 2 days, which allows once-a-day dosing [29].

### 2.5. Behavioral Tests

Behavioral tests including the thermal sensitivity, open field, Y-maze, T-maze, and novel object recognition were performed to assess response to pain, anxiety-like behavior, exploration, spatial reference memory, cognitive ability, and working memory, respectively.

#### 2.5.1. Thermal Sensitivity Test

Rats were placed in transparent plexiglass boxes and were allowed a minimum of 30 min before starting each session for familiarization with the environment. A nociceptive radiant heat spot (50 °C) was projected from a 160-watt light bulb to the shaved lower abdominal surface of the rat and was only applied when the rats were stationary. The rats’ abdomens were shaved a day before the test. Each rat was tested twice per session, separated by a minimum interval of 5 min. The measurements made on all rats in an experimental group were averaged for the session and expressed as mean ± SEM.

#### 2.5.2. Open Field

The open-field test was used to assess spontaneous locomotor activity and detect potential signs of anxiety-like behavior induced by UTI [30]. The experiment was performed in dim light settings to boost further exploration. The rats were allowed 20 to 30 min to habituate to the experimental room and then were successively placed in the open field apparatus (no prior exposure to the apparatus) for 5 min of testing that was recorded by a camera. The videos were analyzed using Any Maze^™^ software (Fort Worth, TX, USA) and variables such as number of entries to central zone, total time spent in the central zone of open field, total distance traveled, average speed, and total time spent immobile was recorded. The central zone is designated as 25% of the area of the open field apparatus [30,31].

#### 2.5.3. Novel-Object-Recognition (NOR)

The novel object recognition test was performed in the Open Field apparatus and on the second day after the rats have performed the open field test and became familiar with the apparatus. The test was performed on day 3 post infection and consisted of 2 phases: a familiarization phase and a testing phase. In the familiarization phase, two identical objects were placed in 2 adjacent corners of the open field, then rats were placed in the open field facing the opposite side to the objects and allowed to explore for 5 min. In the testing phase, one of the objects was replaced by a different novel object (location was kept the same), and the animals were allowed to explore for 5 min in the presence of the novel object. The inter-phase separation time is 5 min, in which the animal is briefly returned to the cage and the apparatus is wiped with 70% ethanol. The number of times the rat visits the area of the novel object and the familiar object and the total time spent there were recorded and analyzed on AnyMaze^™^ software (Fort Worth, TX, USA).

#### 2.5.4. Y-Maze Test

This test was used to assess novel arm exploration in the different groups. The apparatus consisted of three identical arms (10 cm wide and 40 cm long) that are equally spaced (120° apart). No intra-maze cues were added, but different objects were placed at a range of distances outside the maze that would be visible to the rats and serve as extra-maze cues. Training and testing were performed as previously described [32]. Briefly, the test mainly consisted of two phases that were 1 h apart. In the first training or acquisition phase, one of the arms denoted as the novel arm was blocked and the rats were placed in the “start” arm. A period of 10 min was timed for the rats to explore and familiarize themselves with both the “start” (S) and the “familiar” arm (F). In the second phase or the test phase (retention trial), the closed/novel arm (N) was opened, and rats were also placed in the “S” arm. The rats were allowed to roam and explore the three arms for 5 min and videos were recorded on camera and analyzed using Any Maze^™^ software (TX, USA), whereby the number of entries and the total time spent in each arm were recorded. The floor and walls of the maze were wiped with 70% alcohol at the end of the trial with each rat to avoid odor cues.

#### 2.5.5. T-Maze Test

The T-shaped maze had three arms (90° apart); one denoted as a start arm and the two side arms were denoted as the “goal arms”. This test for spontaneous alteration was carried out as previously described by Deacon and Rawlins [33]. The animals were first put in the maze in the start arm facing the maze wall opposite to the goal arms. Once the rat chooses and enters a goal arm, it was locked in that arm for 30 s. After the 30 s, the lock was lifted, and animals are carried gently and placed in the start arm again facing the wall, and then the rats’ goal arm of choice was recorded again. An entry in an arm was denoted as full entry of the animal along with its tail. The rats’ natural tendency in a T-maze was to alternate their choice of goal arm. There was a central partition between the goal arms that was placed to produce more reliability in alteration rates. If the rat chose the goal arm that it was locked in, then it failed to spontaneously alternate and if it chose the opposite arm where it was not locked, then it was recorded as a spontaneous alteration. The response on each trial varied according to what the rats had previously chosen. This procedure was repeated four times per rat and the percentage of spontaneous alterations per rat in total was calculated and the average was calculated per group.

### 2.6. BrdU Injections

BrdU powder (5′-bromo-2-deoxyuridine, Sigma-Aldrich, St. Louis, MO, USA) was dissolved in 0.9% warm sterile saline and was given over three injections (66 mg/Kg/300 μL injection, i.p.) on day 3 post infection. The dosage, volume, and time of injection were based on a protocol followed in a previous work by our group [34].

### 2.7. Sacrifice and Tissue Collection

Rats were deeply anesthetized by i.p. injection of ketamine (Ketalar^®^; 50 mg/kg) and Xylazine (Xylazine^®^; 12 mg/Kg). Bladder and urethra were collected in cryovials under sterile conditions, snap frozen in liquid nitrogen, then transferred to −80 °C for later processing and protein extraction. After that, rats were perfused transcardially with a solution of 0.9% saline followed by 4% formalin. The brains were carefully removed and fixed in 4% paraformaldehyde for 24 h before being transferred to 30% sucrose solution in 0.1M PBS to be stored at 4 °C until full impregnation. Brain sections were cut using a sliding microtome and sampled in a systemic manner as 6 sets using the fractionator method as previously described [32,35]. In brief, 40 μm coronal sections were cut serially, from the rostral to the caudal extent of the DG at the following rostro-caudal coordinates covering the whole hippocampal formation (−2.12 to −6.3 mm relative to bregma). To highlight the topographic distribution of BrdU-positive cells, the DG region was divided into three areas as follows: rostral ranging from −2.12 to −3.7 mm relative to bregma, intermediate ranging from −3.7 to −4.9, and caudal ranging from −4.9 to −6.3 [28,36]. All the sections were collected and stored in sodium azide solution (15 mM in 0.1 M PBS).

For groups sacrificed at 6 and 24 h post infection, aorta excision was performed under anesthesia with no perfusion-fixation. Fresh bladder and urethra tissues were collected on ice as with previous groups. In addition, the brain was extracted, and the hippocampi were cut on ice, snap frozen with liquid nitrogen, and stored at −80 °C for later RNA extraction.

### 2.8. Immunofluorescence Assay

The tissues were stained by a NSC marker and mature neuronal marker; BrdU and NeuN, respectively. Wells containing rostral, intermediate, and caudal dentate gyrus regions for each rat were chosen randomly and immunofluorescence was performed as previously described by Chamaa et al. [34]. Briefly, sections were washed and incubated at 37 °C with 2N HCL to allow denaturing of DNA, then 0.1M sodium borate (pH 8.5) was added for 10 min. The tissues were incubated overnight with monoclonal primary antibodies: mouse anti-BrdU (1:500; Santa Cruz) and rabbit anti- NeuN (1:500; Millipore) diluted in PBS with 3% NGS, 3% BSA, 0.1% Triton-X. Incubation with secondary antibodies would follow on the second day, for 2 h on a shaker at room temperature, using Alexa Fluor-568 goat anti-mouse (1:250; Molecular Probes, Invitrogen) and Alexa Fluor-488 goat anti-rabbit (1:250; Molecular Probes, Invitrogen). Finally, sections were washed and mounted onto slides with Fluoro-Gel with DAPI (Electron Microscopy Sciences, USA). For IBA-1 and GFAP staining, staining was performed as described above, except that sections were directly washed and placed in 10% blocking solution. For IBA-1, rabbit anti-IBA-1 primary antibody (1:1000; WAKO) and Alexa Fluor-568 goat anti-rabbit secondary antibody were used (1:250; Molecular Probes, Invitrogen). For GFAP, rabbit anti-GFAP primary antibody (1:1000; Abcam) and Alexa Fluor-568 goat anti-rabbit secondary antibody were used (1:250; Molecular Probes, Invitrogen).

### 2.9. Cell Counting and Confocal Microscopy

The counting of BrdU^+ve^ cells is strictly confined to the subgranular zone (SGZ) of the DG on day 1 post BrdU injection (proliferation) and to the granular cell layer (GCL) on day 32 post BrdU injection (Neurogenesis). BrdU^+ve^ cells were counted manually using 40×-oil objective. The total number of positive cells counted per rat was multiplied by 6 (the number of sets per rat), to denote the overall number of BrdU^+ve^ cells in each region (rostral, intermediate, and caudal) of the DG per rat. For consistency, BrdU^+ve^ cells were counted by the same researcher and images were acquired under the same laser and microscopic parameters.

Z-stack and tile scan images were taken using a Zeiss LSM 710 confocal microscope at the 40-X oil objective. Z-stacks were used to show all BrdU+ cells distributed within the 40 µm section in the whole dentate gyrus of each region. The images were analyzed using Zeiss ZEN 2009 image-analysis software (Baden-Württemberg, Germany) and were processed with maximal intensity projection.

Representative immunofluorescent images for IBA-1 and GFAP were captured using laser screening on the confocal microscope. Per brain section, 3 snap shots covering and spanning the DG region were quantified and averaged for signal intensity per section. Per rat, signal intensity was quantified in 5 sections taken from the same region for consistency (intermediate DG) and their average was taken to represent signal intensity for each rat. Signal intensity was represented as an average of values from 9 rats per group.

### 2.10. Enzyme-Linked Immunosorbent Assay (ELISA) Assay

Supernatants of the homogenized tissues are used for the detection of IL-1β and IL-8. The protein concentration of each sample is first determined using the Lowrey protein assay according to the manufacturer’s instructions (Bio-Rad Laboratories, Hercules, CA, USA). Pro-inflammatory cytokine release in response to infection was measured using a two-site sandwich ELISA assay. Pro-inflammatory cytokines: interleukin 8 (IL-8) and interleukin 1beta (IL-1β) were screened in protein extracts of urethra and bladder tissues. ELISA plates were prepared according to the manufacturer’s instructions in a four-day protocol as described previously [37,38]. Nunc 96-well immuno plates were coated with immunoaffinity-purified polyclonal sheep anti-rat IL-1β or IL-8 coating antibodies (100 μL/well; NISBC, South Mimms, UK) diluted in coating buffer and kept overnight at 4 °C. The next day, three washes were performed, followed by one-hour incubation with blocking solution (3% BSA, 0.1% Tween20 in PBS) at 37 °C. After three washes, samples and standards were added in duplicates to respective wells to be incubated overnight at 4 °C. On day 3, respective biotin-conjugated immunoaffinity-purified polyclonal antibody (1:4000; 100 μL/well; NISBC, UK) diluted in wash buffer containing 1% Normal Sheep Serum (NSS; abcam, Cambridge, UK) were added after three washes and incubated at 4 °C overnight. On day 4 of the protocol, three washes were performed and then the streptavidin horseradish peroxidase enzyme (Amersham; diluted 1:8000) diluted in wash buffer with 1% BSA was added for 30 min in a shaker. Finally, the samples were incubated with the substrate, tetramethylbenzidine (TMB), along with H_2_O_2_ for 15 min before the reaction was stopped with the sulfuric acid solution (1 M H_2_SO_4_; 100 μL/well). Absorbance intensity was measured using the microplate ELISA reader (Multiscan EX) at 450 nm. A four parameter logistic curve-fit on Prism 7 GraphPad package (GraphPad software, Inc., San Diego, CA, USA) was used for obtaining a standard curve to interpolate samples’ concentrations. Cytokine levels were expressed as picograms per milligram protein.

### 2.11. Conventional PCR

The *E. coli* 1176 clinical isolate was positive for New Delhi metallo-β-lactamase variant 5 (*NDM*-*5*) (Appendix A). Conventional PCR was performed to confirm and monitor the presence of the *E. coli* 1176 bacterial isolate in urine samples over time through the detection of NDM-5 presence. The positive control used in experiments is the extracted DNA of clinical isolate 1176. The primer sequence is in Table 2.

### 2.12. RNA Extraction and Quantitative Real-Time PCR

RNA was extracted from hippocampi tissues using TriZol (TRI reagent^®^, Sigma; St. Louis, MO, USA) and following the manufacturer’s protocol for RNA extraction from tissues. In brief, 1 mL of TriZol reagent was added to tissues gradually and the tissues were homogenized on ice, then 0.2 mL chloroform was added, followed by centrifugation at 12,000 rpm for 20 min at 4 °C. The isolated RNA phase was mixed with 0.35 mL isopropanol, incubated for 10 min at room temperature, centrifuged at 15,000 rpm for 30 min at 4 °C, washed twice with 70% ethanol, and then the pellet was left to air dry before suspending it in RNase free H2O. List of primers used and their sequences are listed in Table 3. cDNA synthesis was performed using a QIAGEN QuantiTect reverse transcription kit following the manufacturer’s protocol. cDNA was diluted in a 1:10 volume ratio. Concentrations and integrity (RNA integrity number—RIN) of isolated RNA were determined using ThermoScientific^TM^ NanoDrop 2000^TM^ and Agilent BioAnalyzer 2100^TM^, respectively. The mRNA expression of sham and UTI hippocampi samples were analyzed by RT-PCR (Bio-rad CFX^TM^ Manager Software; cat #1845000; Hercules, CA, USA) using the ΔΔC_t_ method and the SYBR green system (Applied Biosystems; cat #A46111; Waltham, MA, USA). The PCR reaction consisted of a DNA denaturation step at 95 °C for 5 min, followed by 40 cycles (denaturation at 95 °C for 10 s), then annealing at the appropriate temperature of 57 °C for each primer for 30 s, and finally an extension step at 72 °C for 10 min. For each experiment, reactions were performed in duplicates and the expression of individual genes was normalized to the house keeping gene *Gapdh*. Gene expression was calculated through the following equation:ΔΔC_t_ = ΔC_t (target)_ − Average [ΔC_t (Sham)_](2)
where ΔC_t_ = C_t (target)_ − C_t (GAPDH)_. The amount of endogenous target gene relative to a calibrator (*GAPDH*) became 2^−ΔΔCt^.

### 2.13. Statistical Analysis

Statistical analysis and plotting of figures were made using the Prism 7 GraphPad package (GraphPad software, Inc., San Diego, CA, USA). An unpaired Student’s *t*-test was used to assess the statistical significance of difference between the two groups; sham rats and rats with a UTI across the following parameters: number of BrdU-positive cells, number of BrdU/NeuN double-positive cells, parameters in the open field, concentration of cytokines on day 4 post infection, time spent in each arm of the y-maze, latency to enter the novel arm, time spent exploring the object in the novel object recognition test, and the optical intensity of IBA-1 and GFAP. One-way ANOVA followed by Tukey’s multiple comparison was used to test the statistical significance whenever comparing a variable across three groups in the study. The measure of statistical significance for IL-1β and IL-8 concentrations in sham rats and rats with a UTI sacrificed at 6 or at 24 h was analyzed by one-way ANOVA followed by Tukey’s multiple comparison test. Moreover, one-way ANOVA followed by Tukey’s multiple comparison test was also used to assess statistical significance for the number of BrdU-positive cells between sham rats, rats with a UTI, and rats with a UTI treated with Fosfomycin or Piroxicam. All data were averaged per group and presented as mean ± standard error mean (SEM). The *p* value of <0.05 was considered as the limit of significance of differences at a 95% confidence interval.

## 3. Results

### 3.1. Urine Infection and Increased Levels of Cytokines Following UTI

Rats included in the study were confirmed to have UTI as the CFU counts in their urine exceeded 10^5^ CFU/mL on day 1 post infection (Figure 2A,B) and by detection of bla_NDM-5_ in their urine culture (Figure 2C,D). Moreover, infected rats had increased levels of pro-inflammatory cytokines production in their bladder and urethra as compared to vehicle-treated sham rats (Figure 3).

A tendency for the increase in the production of IL-1β was detected 6 h post infection (52.29 pg/mg ± 6.05), which continued to reaching a significant peak at 24 h post infection (81.06 pg/mg ± 19.43, *p* = 0.0315) as compared to non-infected sham rats (32.03 pg/mg ± 2.54) (F(2,12) = 4.332, *p* = 0.0384). Similarly, a trend of increases in the production of IL-8 was detected at 6 h post infection (86.31 pg/mg ± 15.93), which later significantly peaked at 24 h post infection (202.21 pg/mg ± 57.96, *p* = 0.0003) as compared to sham non-infected rats (26.81 pg/mg ± 2.38) (F(2,12) = 17.05, *p* = 0.0003). At day 4 post infection, the increase in the production of IL-1β in urethra was sustained (42.76 pg/mg ± 5.9, *p* = 0.036, unpaired Student’s *t*-test) and was significantly higher than that of sham non-infected rats (26 pg/mg ± 3.93). However, the levels of IL-8 in urethra at day 4 post infection (23.7 pg/mg ± 5.4) subsided to sham levels (10.02 pg/mg ± 2.98) (Figure 3A,B).

In bladder tissues, we detected a significant peak in the production of IL-1β only on day 4 post infection (108.38 pg/mg ± 7.84, *p* < 0.0001) as compared to the sham group (36.16 pg/mg ± 6.35). There were no significant peaks detected at 6 (49.08 pg/mg ± 5.89) and 24 h (37.8 pg/mg± 4.38) as compared to the sham group (48.55 pg/mg ± 6.95) (F(2,10) = 1.431, *p* = 0.2841) (Figure 3C,D). Similarly, this was also consistent with IL-8 production in the bladder, where a significant increase in IL-8 production was only detected on day 4 post infection (80.92 pg/mg ± 4.98, *p* < 0.0001) as compared to the sham rats (11.07 pg/mg ± 1.22). Moreover, there was a slight non-significant increase in IL-8 at 24 h post infection (17.95 pg/mg ± 8.2), as compared to IL-8 concentrations at 6 h (5.45 pg/mg ± 0.69) and the sham group (5.57 pg/mg ± 0.9) (Figure 3C,D).

### 3.2. UTI Decreased Proliferation of Neural Stem Cells in the DG at Four Days Post Infection

NSCs were stained with BrdU, while mature dentate gyrus granular cells were stained with NeuN. The purpose of this double staining was to allow localization and counting of the newly born BrdU-positive cells only in the sub-granular zone. Since rats were sacrificed 24 h post BrdU injection, as expected, the BrdU-positive cells were not co-stained with the mature neuronal marker; NeuN. Rats with confirmed UTI had a significantly decreased number of BrdU-positive cells (2452 ± 243; *p* < 0.001, unpaired Student’s *t*-test) in their DG as compared to sham rats (4544 ± 303) at day 4 post infection (Figure 4A,B).

Topographically, the decrease in BrdU-positive cells was significantly noted in all regions of the DG of rats with a UTI; rostral DG (343 ± 93 versus 845 ± 189; *p* = 0.03), intermediate DG (413 ± 73 versus 757 ± 68; *p* = 0.0033), and caudal DG (1684 ± 187 versus 2943 ± 298; *p* = 0.0025), as compared to sham rats, respectively (unpaired Student’s *t*-test) (Figure 5A).

### 3.3. UTI Decreased Neurogenesis in the DG at 34 Days Post Infection

As neurogenesis takes around 30 to 40 days, some of the BrdU-positive NSCs that were born at time of BrdU injection have become mature NeuN-positive neurons. Total BrdU-positive cells were counted to a total count of BrdU-positive cells present at the time of sacrifice. Rats with confirmed UTI had a decreased number of BrdU-positive cells (829 ± 101 cells; *p* = 0.04) as compared to that in the DG of sham rats (1477 ± 280 cells). It is expected that a number of the newly born cells at time of BrdU injection would die as a normal part of neuronal turnover and the number would differentiate into either astrocytes or oligodendrocytes. Thus, the number of BrdU/NeuN double-positive cells were counted to assess the number of BrdU-positive cells that have specifically differentiated into neurons. The number of BrdU/NeuN double-positive cells in the DG rats with a UTI (764 ± 90 cells; *p* = 0.03) was significantly lower than that of sham rats (1374 ± 248 cells) (Figure 6A,B,D). Both the sham group (93 ± 1%) and UTI rats (93 ± 2%) had comparable percentages of BrdU/NeuN double-positive cells out of the total BrdU-positive-labelled cells (Figure 6C). Thus, while the survival of NSCs was affected in the DG of rats with a UTI, the differentiation of the surviving NSCs into neuronal fate was not affected.

Topographically in the DG, the decrease in BrdU-positive cells was significantly notable in the intermediate (152 ± 23; *p* = 0.04) and caudal DG (434 ± 50; *p* = 0.02) of rats with a UTI as compared to sham rats (292 ± 59 and 821 ± 139, respectively) (unpaired Student’s *t*-test). As for the rostral DG, the total number of BrdU-positive cells was comparable between sham (365 ± 89) and UTI rats (233 ± 41) (Figure 5B).

### 3.4. Treatment with the Antibiotic Drug Fosfomycin Decreased Basal Levels of the Proliferation of NSCs

Fosfomycin injection cleared the infection after one day of treatment as seen with the CFU assay (Appendix A). Both the sham group (1556 ± 120; *p* < 0.0001) and rats with a UTI (1092 ± 204; *p* < 0.0001) treated with Fosfomycin had significantly decreased numbers of BrdU-positive cells as compared to untreated sham rats (4544 ± 303) (F (3,24) = 33.88; *p* < 0.0001) (Figure 7A,B).

### 3.5. Treatment with the Anti-Inflammatory Drug Piroxicam Did Not Alter the Number of NSCs in the Sham Group and Rats with a UTI

Treatment with the NSAID Piroxicam did not induce significant alteration in the number of BrdU-positive cells in sham rats (Figure 8B). Sham rats treated with Piroxicam (3790± 198) still had a significantly higher number of BrdU-positive cells as compared to untreated rats with a UTI (2452 ± 243; *p* = 0.0154). Rats with a UTI that were treated with Piroxicam had a comparable number of BrdU-positive cells to untreated rats with a UTI (2140 ± 243 versus 2452 ± 243, respectively). Moreover, the number of BrdU-positive cells in rats with a UTI treated with Piroxicam (2140 ± 243) was significantly lower than the number of BrdU-positive cells in the sham group treated with Piroxicam (3790 ± 198; *p* = 0.0078) and in untreated sham rats (4544 ± 303; *p* < 0.0001) (F(3,24) = 17.7; *p* < 0.0001) (Figure 8A,B).

### 3.6. UTI Elevated the mRNA Expression of Il-1β and Decreased That of Bdnf, Ngf, and Fgf2

There was a significant increase in the levels of *Il*-*1**β* mRNA detected at 6 h post infection in the hippocampi of rats with a UTI (1.97 ± 0.28, *p* = 0.025, unpaired Student’s *t*-test) as compared to mRNA levels in the hippocampi of sham rats (1.02 ± 0.12). This increase returned to sham-levels at 24 h post infection (2.41 ± 1.09) (Figure 9A). There were no significant changes in the mRNA levels of *Il*-*6* in hippocampi of rats with a UTI at 6 h (1.04 ± 0.2) and 24 h (1.67 ± 0.55) as compared to sham rats (1.05 ± 0.18) (Figure 9B).

On the other hand, mRNA levels of *Bdnf*, *Ngf*, and *Fgf* were significantly lower in the hippocampi of rats with a UTI at 6 h post infection (0.36 ± 0.17, *p* = 0.034; 0.45 ± 0.17, *p* = 0.049; and 0.6 ± 0.07, *p* = 0.007, respectively) as compared to expression level in sham hippocampi (1.03 ± 0.13, 1.04 ± 0.15, and 1.22 ± 0.11, respectively). Basal levels were recovered at 24 h post infection (0.78 ± 0.15, 1.02 ± 0.13, and 0.95 ± 0.12, respectively) (one-way Anova followed by Tukey’s multiple comparison test, F(2,11) = 4.469, *p* = 0.038 for *Bdnf*; F(2,10) = 5.066, *p* = 0.0302 for *Ngf*; F(2,10) = 8.253, *p* = 0.0076 for *Fgf2*) (Figure 9C–E).

### 3.7. UTI Did Not Induce Significant Changes in Microglial and Astrocytic Cells

The signal intensity of microglial cells stained with IBA-1 in the DG was comparable between rats with a UTI (23.78 A.U. ± 1.36) and sham rats (23.6 A.U. ± 1.33) (Figure 10A,C).

Similarly, the signal intensity for GFAP staining astrocytic cells in the DG was also comparable between rats with a UTI (23.32 A.U. ± 1.08) and sham rats (24.7 A.U. ± 1.45) (Figure 10B,D).

### 3.8. Increased Heat Sensitivity in Urinary Tract Infected Rats

Rats with a UTI displayed a shorter latency of withdrawal reflex in reaction to nociceptive heat (10.45 s ± 1.69) as compared to vehicle-treated sham rats (18.84 s ± 2.62; *p* = 0.0175, unpaired Student’s *t*-test). This decrease in latency denotes a heat hyperalgesia induced at the level of the abdominal skin of rats with a UTI (Appendix A).

Treatment with Fosfomycin induced a recovery of the latency of withdrawal reflex in rats with a UTI (17.86 s ± 3.63) to that observed in sham rats (18.84 s ± 2.62) (Appendix A).

Moreover, Piroxicam treatment abolished the heat hyperalgesia observed in rats with a UTI, as assessed by the return of the latency of the withdrawal reflex to a level comparable to that observed in sham rats (18.06 s ± 3.99 versus 18.84 s ± 2.62) (Appendix A).

### 3.9. Rats with a UTI Displayed Normal Spontaneous Locomotor Activity and Exploratory Behavior

Rats with a UTI recorded comparable measurements for the following indicators as compared to sham rats, respectively: total time spent in the central zone (27.63 s ± 4.66 in UTI versus 23.72 s± 5.14 in sham), latency to enter central zone (18.8 s ± 4.36 versus 15.22 s ±4.56), average speed (0.036 m/s± 0.003 versus 0.038 m/s± 0.0034), total distance traveled (10.67 m ±1 versus 11.59 m ± 1.01), total mobility time (150.39 s± 15.36 versus 184.07 s± 12.98), and total immobility time (149.69 s ± 15.36 versus 115.93 ± 12.98 s) (Appendix A).

### 3.10. Rats with a UTI Had a Similar Tendency to Explore a Novel Object to Sham Rats

Rats with a UTI spent a comparable time exploring the novel object (59.44 s ± 6.58) to the sham rats (75.73 s ± 10.46) in the novel object recognition test. Similarly, the total time spent exploring the familiar object by rats with a UTI (63.81 s ± 7.81) was comparable to sham rats (54.63 s ± 10.12) (Appendix A).

### 3.11. Rats with a UTI Spent Less Time Exploring the Novel Arm in the Y-Maze Test

On day 2 post infection, the total time spent by rats with a UTI in the novel arm (51 s ± 5, *p* = 0.0245) was significantly less than that scored by sham rats (84 s ± 11.65) (unpaired Student’s *t*-test). On the other hand, the total time spent by rats with a UTI in the start arm (139 s ± 20) and familiar arm (97 ± 23) was comparable to that spent by sham rats (116 s ±18 and 86 s ± 11, respectively) (Figure 11A). Moreover, rats with a UTI spent a significantly longer time to first enter the novel arm (42 s ± 10, *p* = 0.0358) as compared to the sham rats (13 s ± 6) (unpaired Student’s *t*-test) (Figure 11B).

On day 34 post infection, the total time spent by rats with a UTI in the start arm (134 s ± 31), familiar arm (87 s ± 21), and novel arm (68 s ± 19) were all comparable to that spent by sham rats (114 s ± 17, 82 s ± 11, and 85 s ± 16, respectively) (Figure 11C).

### 3.12. Rats with a UTI Had Less Tendency to Spontaneously Alternate in the T-Maze Test

Rats with a UTI recorded a balanced alternation rate of 55% ± 12 (*p* = 0.05, unpaired Student’s *t*-test), which is significantly lower than that recorded by sham rats 81% ± 6 (Figure 11D).

## 4. Discussion

In this study, we hypothesized that urinary tract infections could influence brain homeostasis, in particular, hippocampal neurogenesis based on clinical observation of the non-conventional symptoms of UTIs in elderly patients such as confusion and delirium.

Our findings show that UTIs cause a decrease in the proliferation of BrdU-positive neural stem cells in the dentate gyrus of the hippocampus. This effect is further manifested a month post infection as a decrease in the number of BrdU/NeuN double-positive cells and thus decreased adult hippocampal neurogenesis. This finding is in line with accumulated evidence showing a causal link between decreased neurogenesis and neuroinflammation [4,5,12,13] or peripheral inflammation in general [10,39]. In addition to decreased proliferation of NSCs, we also detected a decrease in the mRNA expression of *Bdnf*, *Ngf*, and *Fgf2* in the hippocampus of rats with a UTI. BDNF and NGF are both neurotrophic factors that are well characterized for their critical role in learning and memory [40,41]. FGF2 is a potent multi-functional growth factor implicated in the regulation of neural stem/progenitor cells in the neurogenic niche of the dentate gyrus [42]. In addition, FGF2 levels are also associated with spatial memory learning [43]. On the other hand, an increase in the mRNA expression of *Il*-*1β*, but not *Il*-*6*, was detected in the hippocampi of rats with a UTI. IL-1β was synthesized and released by both neurons and glial cells of the brain. Traditionally considered as pro-inflammatory, this cytokine is not only involved in inflammatory pathways, but also has a well-established role in the brain as a neuromodulator [44,45,46]. IL-1β is required for neuronal differentiation and the normal regulation of hippocampal plasticity and memory [47,48]. However, increased production of IL-1β in the brain, especially the hippocampus, has been previously reported to impair hippocampal-dependent learning and synaptic plasticity [49,50,51]. It is even reported that increased IL-1β induces cognitive decline in Alzheimer’s disease patients [52]. In conclusion, both the decreased expression of *Bdnf*, *Ngf*, and *Fgf* and the increased expression of *Il*-*1β* can be considered as key players in the observed reduction of neurogenesis. Furthermore, glial cells constitute the first potential contributor to the observed alteration in the expression of neurotrophic factors and cytokines. Looking into the signal intensity of microglia and astrocytes, we did not detect changes between rats with a UTI and sham rats. However, this does not completely reflect the activity of these cells. There could be changes at the level of transcriptional profile that would not be detected by signal intensity.

All in all, the above-mentioned findings leave an open question on the mediator between UTIs and decreased neurogenesis. Once an infection is initiated in the periphery, both the released cytokines and LPS convey this information to the brain using both humoral and neuronal routes of communication [53]. It could be speculated that this link could be due to neural mechanisms related to the activation of the hypothalamic-pituitary-axis (HPA) due to inflammation and nociception [54]. Moreover, the release of pro-inflammatory cytokines in the periphery activates the HPA, and results in elevated levels of glucocorticoids, which are known to strongly suppress proliferation of NSCs [55]. On the other hand, the hippocampus is emerging as a key brain region involved in the processing of pain as multiple evidence shows altered hippocampal plasticity and cytokine expression in animal models of chronic pain [56,57]. Interestingly, research has shown that pain is associated with suppressed neurogenesis and altered short-term synaptic plasticity in a way similar to stress [58]. In accordance with this, we found that infected rats had a higher sensitivity to thermal pain in their abdominal area overlying the urinary tract. This is in line with previous reports on UTIs and pain response [59,60]. Thus, it is possible that both the activation of the HPA with its consequences and the activation of pain pathways could have mediated the effect seen at the level of NSC proliferation. This interplay between the brain and the peripheral immune system has been the focus of recent studies, including one that shows that the brain’s insular cortex stores immune-related information and carries neuronal representations of inflammatory information [61]. Furthermore, IL-8 is a chemoattractant cytokine that activates the sympathetic nervous system and has the potential to form a bidirectional communication between the nervous system and the immune system [62,63,64]. Thus, IL-8 could be considered as another important mediator between peripheral inflammation and the brain [65].

It is worth noting that the infection did not alter the rats’ normal spontaneous locomotor activity and exploration in an open field test. These findings are not in favor of the hypothesis attributing the impaired performance of rats in tests for exploration and memory due to sickness behavior as a result of the infection. Rats with a UTI seem to have impaired spatial reference memory as shown by their performance in the Y-maze test. We have previously reported similar findings relating levels of NSCs proliferation and neurogenesis to performance in a Y-maze test [32]. By contrast, performance in the novel object preference/recognition test was comparable between rats with a UTI and the sham group, suggesting there is no impairment in the rats’ recognition memory. This could be expected as the novel object recognition task evaluates the rats’ non-spatial learning of an object’s identity, which also involves several brain regions [66]. In addition, rats with a UTI had a considerably lower percentage of spontaneous alteration in the T-maze test, which evaluates the short-term working memory. This is in line with previous reports showing altered performance in the T-maze following peripheral inflammation [67]. Thus, the cellular and molecular changes in the brain were paralleled by changes in cognitive behavior in some memory-performance tasks.

To investigate the possible factors playing a role in the interplay between urinary tract infections and the brain, we opted for two different kinds of treatments: treatment with NSAIDs and treatment with antibiotics. Treating the rats with Fosfomycin was able to clear the bacterial infection (Appendix A), on one hand; however, it reduced basal levels of hippocampal NSCs proliferation on the other hand. It has been previously shown that Fosfomycin penetrates the cerebrospinal fluid (CSF) even in the presence of an intact blood-brain barrier (BBB) and is even suggested for the treatment of CNS infections [68,69]. This could explain why the use of this drug had an effect on hippocampal NSCs’ proliferation. Nonetheless, treatment with Fosfomycin recovered pain sensitivity at an early time point post infection. It should be noted that such effects of Fosfomycin at the level of the brain have not been previously reported or investigated in the literature. Thus, here, it is difficult to draw a conclusion on whether treatment with antibiotics could have recovered NSCs’ proliferation since Fosfomycin by itself further decreased their proliferation.

The effect of NSAIDs has been recently investigated in clinical trials for the treatment of UTIs and has yielded variable results from recovery to worsening of symptoms [70,71]. We used the NSAID, Piroxicam, as we previously established that it does not affect the basal levels of hippocampal neurogenesis [5]. Treatment with Piroxicam did not reverse the reduction in proliferation of NSCs in rats with a UTI, nor did it affect the basal levels of proliferation in sham rats. On the other hand, piroxicam treatment recovered hyperalgesia in rats with a UTI. The prophylactic administration of Piroxicam might have exacerbated the infection with time instead. However, it is beyond the scope of our study to investigate further treatment regimens involving NSAIDs. Further experiments that use LPS intra-urethral injection would be beneficial in investigating peripheral inflammation in the urinary tract in the absence of bacterial infection.

## 5. Conclusions

In conclusion, we present novel findings on the effect of urinary tract infections on hippocampal neurogenesis and cognitive behavior. This could, at least, partly explain symptoms of confusion and delirium seen during episodes of UTI, especially in elderly patients. Peripheral inflammation is a suspected factor for changes seen at the level of the brain. It has been previously reported that peripheral inflammation, in particular toll-like receptor-induced inflammation, induces remote global gene expression changes in the brain [72]. However, the exact mechanism for the effect of peripheral inflammation on the brain requires further investigation. The findings of this study might lead to considering neurogenesis as a potential target in the dynamic crosstalk between the infected urinary tract and the brain.

## Figures and Tables

**Figure 1 biology-11-00891-f001:**
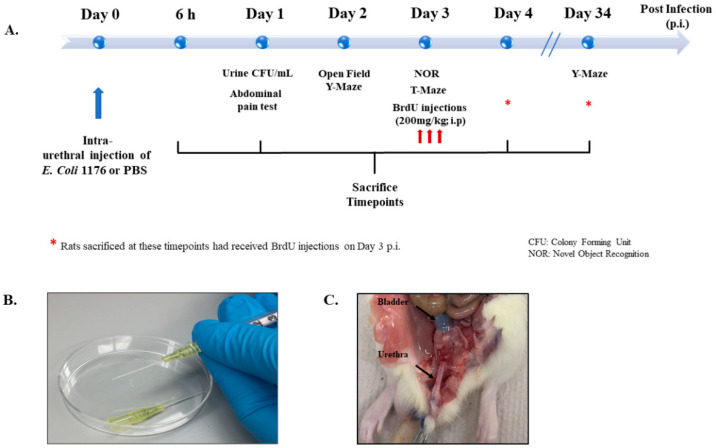
Experimental design and settings. (**A**) Schematic showing the experimental timeline followed for behavioral tests, BrdU injections, and sacrifice. (**B**) Image showing the catheters and syringes used for intraurethral injections in the study. (**C**) Image showing the anatomical location of trypan blue intraurethral instillation in the bladder of a rat used for trial.

**Figure 2 biology-11-00891-f002:**
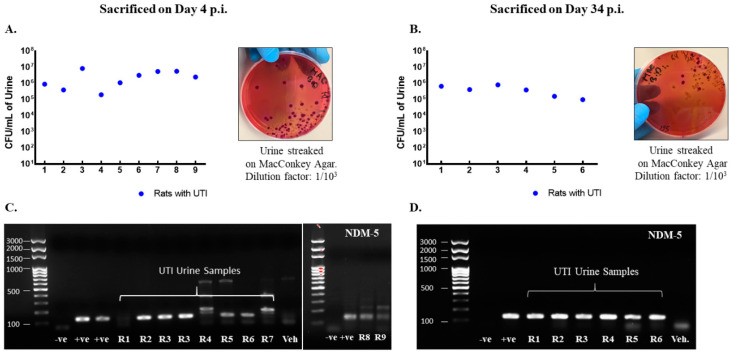
Rats with a UTI confirmed for infection by the presence of NDM−5 bacterial-specific gene in urine and CFU/mL > 10^5^. CFU/mL of urine samples collected on day 1 post infection from rats with a UTI that were either sacrificed on day 4 (*n* = 9) (**A**) or day 34 (*n* = 6) (**B**) post infection along with representative images for the streaked MacConkey agar plates. Each dot represents the CFU/mL value for one rat. All rats had a CFU/mL value above 10^5^ confirming infection. Presence of *Ndm*−*5* gene in urine samples collected on day 1 post infection as confirmed by conventional PCR and shown by gel electrophoresis for rats with a UTI sacrificed on day 4 post infection (**C**) and day 34 post infection (**D**).

**Figure 3 biology-11-00891-f003:**
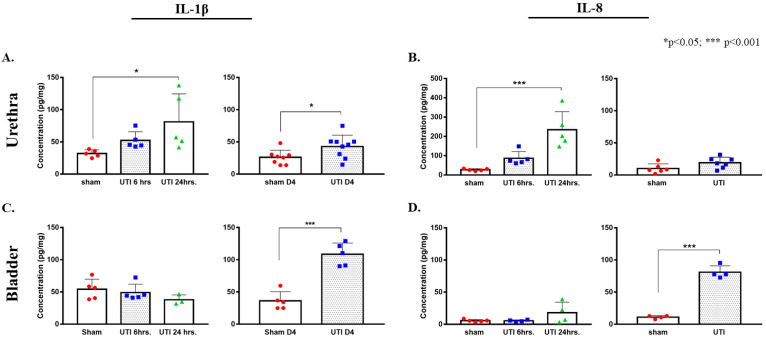
Increased protein concentrations of IL-1β and IL-8 in urethra and bladder tissues in ELISA assay. Concentration of IL-1β (**A**) and IL-8 (**B**) in urethra tissues of rats with a UTI and sham rats at 6 h, 24 h, and day 4 (D4) post infection. Concentration of IL-1β (**C**) and IL-8 (**D**) in bladder tissues of rats with a UTI and sham rats at 6 h, 24 h, and day 4 post infection. Determination of significance between the sham and UTI group at 6 and 24 h was generated using one-way Anova followed by Tukey’s multiple comparison’s test. Determination of significance between the sham and UTI group on day 4 post infection was done using an unpaired Student’s *t*-test.

**Figure 4 biology-11-00891-f004:**
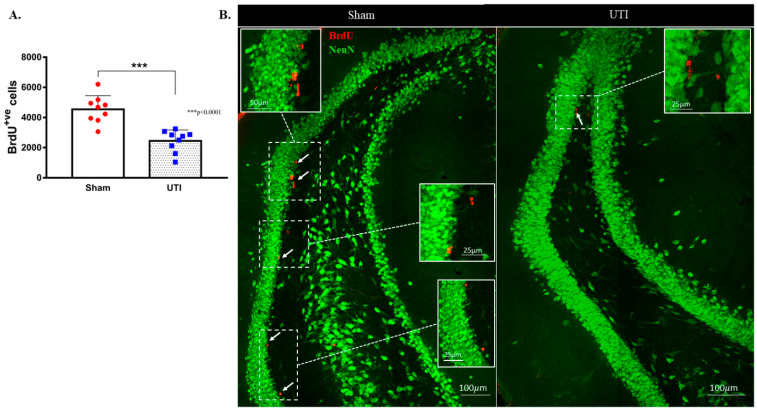
Decrease in the proliferation of DG NSCs on day 4 post infection. (**A**) Stereological quantification of BrdU-labeled cells in the DG of the sham group (*n* = 9) and rats with a UTI (*n* = 9) sacrificed on day 4 post infection. Unpaired Student’s *t*-test was used to determine the statistical significance between the sham (*n* = 9) and UTI (*n* = 9) groups. Each bar represents the average ± SEM of BrdU-positive cells per group and each dot represents the measured number in each rat. This method of presentation was followed for all subsequent figures. (**B**) Representative confocal images showing immunofluorescence labeling of NeuN (green) and BrdU (red; arrows) in the DG of the sham group and rats with a UTI. Images were taken as Z stacks and tile scan using a 40×-oil objective.

**Figure 5 biology-11-00891-f005:**
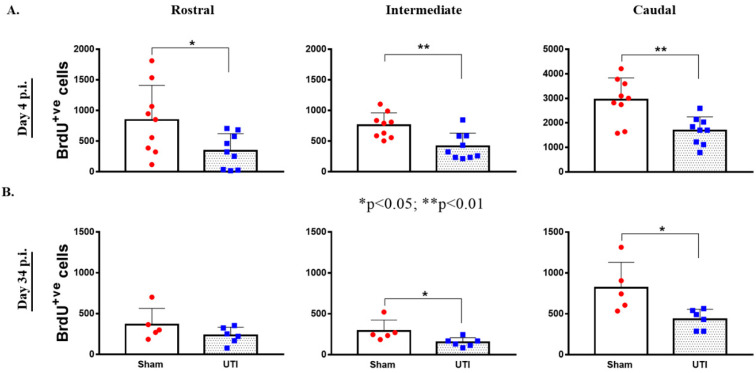
Topographical Distribution of BrdU-positive cells in the DG. Topographical distribution of BrdU-positive cells in the rostral, intermediate, and caudal regions of the DG of rats sacrificed on day 4 (*n* = 9 per group) (**A**) or day 34 (*n* = 5 for the sham group and *n* = 6 for rats with a UTI) post infection (**B**). Determination of the significance of differences was generated using an unpaired Student’s *t*-test.

**Figure 6 biology-11-00891-f006:**
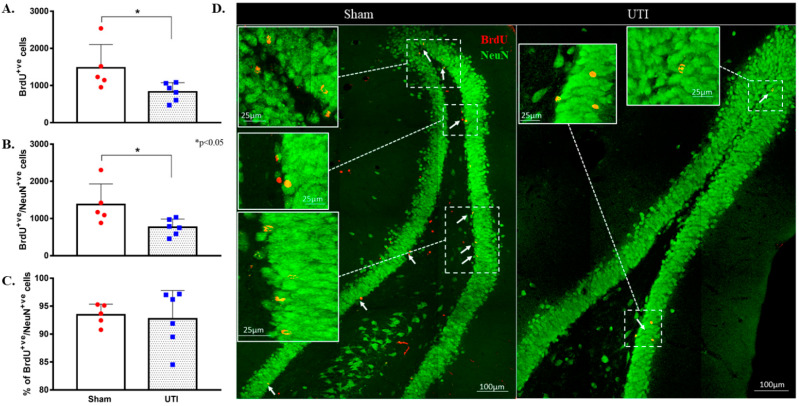
Decrease in neurogenesis persists on day 34 post infection. (**A**) Stereological quantification of the BrdU-labeled cells in the DG of sham rats (*n* = 5) and rats with a UTI (*n* = 6) sacrificed on day 34 post infection. (**B**) Stereological quantification of BrdU/NeuN double-labeled cells in the DG of sham rats and rats with a UTI. (**C**) Percentage of BrdU/NeuN double-labeled cells out of total BrdU-positive cells in the DG of sham and rats with a UTI. An unpaired Student’s *t*-test was used to determine the statistical significance between the sham and UTI groups. (**D**) Representative confocal images showing immunofluorescence labeling of NeuN (green) and BrdU (red; arrows) in the DG of sham rats and rats with a UTI. Images were taken as Z stacks and tile scan using a 40×-oil objective.

**Figure 7 biology-11-00891-f007:**
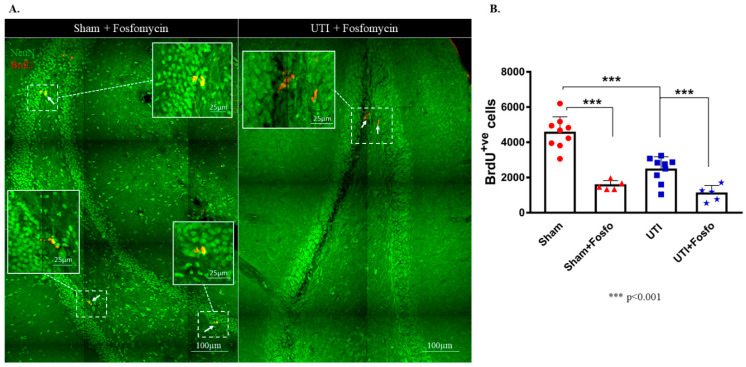
Treatment with Fosfomycin (fosfo) decreases the basal level of BrdU-positive cells and does not revert the decrease in NSCs on day 4 post infection. (**A**) Representative confocal images of the DG of Fosfomycin-treated sham (*n* = 5) and UTI rats (*n* = 5) showing NeuN (green) and BrdU-positive cells (red; arrows). Images were taken as Z stacks and tile scan using a 40×-oil objective. (**B**) Stereological quantification of BrdU-labeled cells in the DG of Fosfomycin-treated sham and UTI rats sacrificed on day 4 post infection. Determination of statistical significance of differences was generated using one-way Anova followed by Tukey’s multiple comparison test.

**Figure 8 biology-11-00891-f008:**
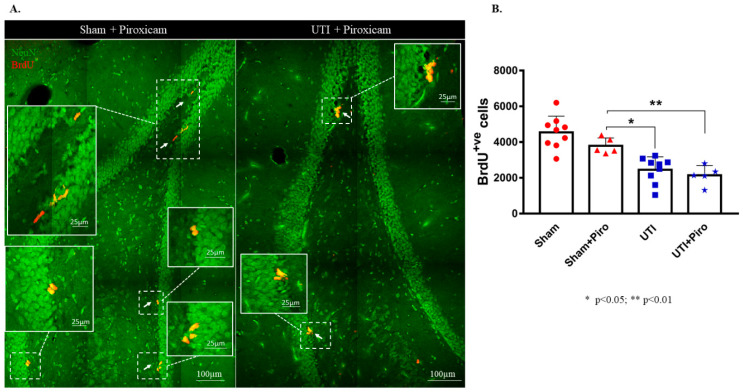
Treatment with Piroxicam (Piro) does not revert the decrease in NSCs on day 4 post infection. (**A**) Representative confocal images of the DG of Piroxicam-treated-vehicle (*n* = 5) and UTI rats (*n* = 5) showing NeuN (green) and BrdU-positive cells (red; arrows). Images were taken as Z stacks and a tile scan using a 40×-oil objective. (**B**) Stereological quantification of BrdU-labeled cells in the DG of Piroxicam-treated sham and UTI rats sacrificed on day 4 post infection. Each bar represents the average ± SEM of BrdU-positive cells per group and each dot represents measured number for each rat. Determination of statistical significance of differences was generated using one-way Anova followed by Tukey’s multiple comparison test.

**Figure 9 biology-11-00891-f009:**
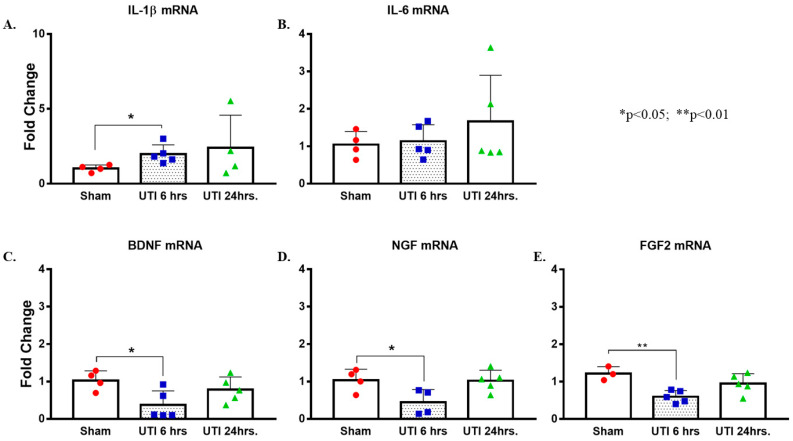
UTI elevated mRNA expression of *Il*-*1β* and decreased that of *Bdnf*, *Ngf,* and *Fgf2*. (**A**,**B**) Increased expression of *Il*-*1β* mRNA with no changes in *Il*-*6* in hippocampi of rats with a UTI versus sham rats, unpaired Student’s *t*-test. (**C**–**E**) Decreased expression of *Bdnf*, *Ngf,* and *Fgf2* mRNA in the hippocampi of rats with a UTI versus sham hippocampi detected at 6 h post infection. One-way ANOVA followed by Tukey’s multiple comparison test, *p* = 0.034, *p* = 0.049, and *p* = 0.007, respectively.

**Figure 10 biology-11-00891-f010:**
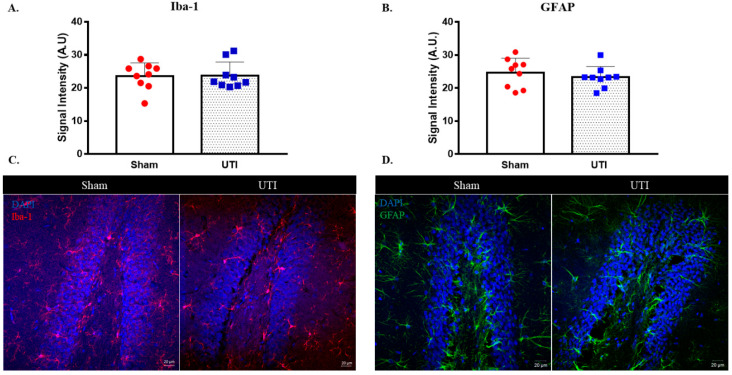
UTI does not induce significant alteration in microglial and astrocytic cells. Signal intensity quantification and representative confocal images for IBA-1 (**A**,**C**) and GFAP (**B**,**D**)-positive cells in the DG of the sham group and rats with a UTI. Signal intensity was quantified using a Zeiss LSM 710 laser scanning confocal microscope. Unpaired Student’s *t*-test was used to determine the statistical significance between the sham and UTI groups.

**Figure 11 biology-11-00891-f011:**
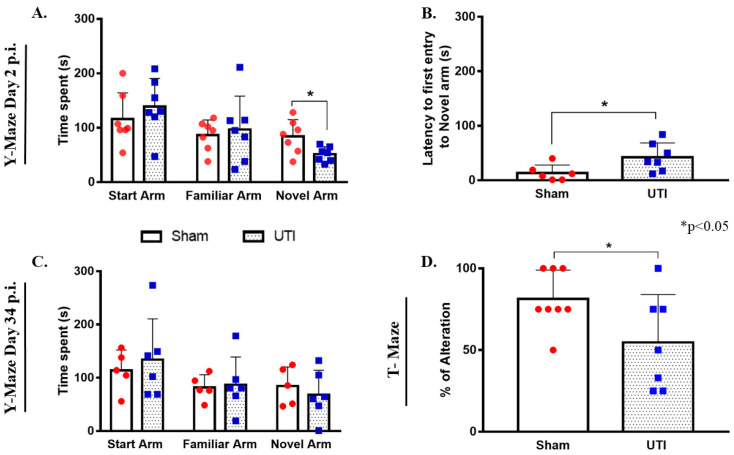
Rats with a UTI spend less time exploring the novel arm in the Y-maze and score lower alternation rates in the T-maze. (**A**) The total time spent in the novel arm by rats with a UTI (*n* = 7) is significantly lower than that spent by sham rats (*n* = 7) in the Y-maze on day 2 post infection. (**B**) Rats with a UTI (*n* = 7) had longer latency to enter the novel arm as compared to the sham group (*n* = 6) in the y-maze performed on day 2 post infection. Statistical significance for latency and time spent in the novel arm between the sham group and rats with a UTI was assessed using an unpaired Student’s *t*-test. (**C**) Rats with a UTI (*n* = 6) spent a comparable time in all arms of the Y-maze as compared to the sham group (*n* = 5) on day 34 post infection. (**D**) Rats with a UTI (*n* = 7) scored a significantly lower percentage of correct alteration in the T-maze test as compared to sham rats (*n* = 8), unpaired Student’s *t*-test.

**Table 1 biology-11-00891-t001:** Experimental groups as per treatments and sacrifice time points.

Time Point of Sacrifice	Groups/Treatments
6 and 24 h post infection	UTI and sham (*n* = 5 each)
Day 4 post infection	UTI and sham (*n* = 9 each) Treatment with Piroxicam; UTI and sham (*n* = 5 each) Treatment with Fosfomycin; UTI and sham (*n* = 5 each)
Day 34 post infection	UTI (*n* = 6) and sham (*n* = 5)

**Table 2 biology-11-00891-t002:** PCR primer for NDM-5 used for detecting the presence of clinical isolate 1176 in urine samples in conventional PCR. F: Forward and R: Reverse.

Target Gene	Conventional PCR Primer Sequence (5′ 3′)
**bla** ** _NDM-5_ **	F: 5′-GGCCAGCAAATGGAAACTGG-3′ R: 5′-CAAACCGTTGGAAGCGACTG-3′

**Table 3 biology-11-00891-t003:** List of primers used with qRT-PCR in the study. Brain-derived neurotrophic factor: *Bdnf*; Glyceraldehyde-3-Phosphate Dehydrogenase: *Gapdh*; Nerve growth factor: *Ngf*; Fibroblast growth factor 2: *Fgf2*.

Rattus Norvegicus Primers	Sequence (5′->3′)	Product Length
** *Gapdh* **	F: TCACCATCTTCCAGGAGCGA R: GGCGGAGATGATGACCCTTT	149
***IL*-*1β***	F: AGGCTGACAGACCCCAAAAG R: GGTCGTCATCATCCCACGAG	264
***IL*-*6***	F: ACAAGTCCGGAGAGGAGACT R: ACAGTGCATCATCGCTGTTC	167
** *Bdnf* **	F: CTCCGCCATGCAATTTCCAC R: CAGCCTTCATGCAACCGAAG	279
** *Ngf* **	F: CATCGCTCTCCTTCACAGAGTT R: TCTGTGTACGGTTCTGCCTG	222
** *Fgf2* **	F: AGGATCCCAAGCGGCTCTAC R: TACCGGTTCGCACACACTC	166

## Data Availability

All data are contained within the article and Appendix A.

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
