# Peer review of "Urinary Tract Infections Impair Adult Hippocampal Neurogenesis"

_biology, 2022, doi:10.3390/biology11060891_

Round 1

Reviewer 1 Report

Darwish and colleagues aimed to investigate the impact of urinary tract infections (UTI) on rat hippocampal neurogenesis. They found that UTI impairs neurogenesis and this is associated with cytokine level differences and some differences in the Y-maze test.  The study is interesting and will be of general interest to the Biology audience. However, there are several concerns that need to be addressed.

  1. Define E. coli in the abstract. As a standard microbiology nomenclature, it should be define first (e.g. Escherichia coli in italics, then, E. coli not E. Coli.
  2. Include the IACUC number from your institution.
  3. Fig. 1 needs to be explained better in the Figure legend (e.g., A, B, C). Also, the Schematic or diagram not Schematic diagram should be clearer.
  4. UTI model methods needs to be summarized and clearly explained.
  5. Results should be described better.
  6. If you define nonsteroidal anti-inflammatory drug (NSAID), the use the abbreviation consistently. Check for others as well.
  7. Define Xylazine.
  8. Describe clearly the tissues used for cytokine determinations in the materials and methods, results, and legends.
  9. Statistics should be clearly explained in 2.13. You don't need to include each statistical test in each result if you explain it cohesively in 2.13. It makes the reading of the manuscript difficult.
  10. Add supplemental Fig. 1 to the results.
  11. Fig. 2 and all the fluorescent figures. Describe the colors (e.g., green, red, etc.) in the legends.
  12. The authors concluded that there are no changes to glial cells. Do they refer to number or morphology? If numbers only, did they check for morphological changes? E.g., number of processes per cell, thickness, distrophic cells, etc. 
  13. The discussion is an iteration of the results. In fact, the first 3 paragraphs are not required. Delete please, this information was already described.
  14. SFig. 2 and 3 can be included in the study.
  15. How do you difference changes in behavior due to inflammatory mediators from microbial components?

Reviewer 2 Report

In the current paper, the authors study the effect of urinary tract infections on hippocampal neurogenesis and cognitive behavior.

Overall the study is well designed and executed, and in general results are described in a logical manner. There are, however, some major and minor concerns listed below.

Major concerns:

  1. Paragraph from line 344 to 372: Different statistical test are used to describe the same graphs, which I do not understand. Some columns are compared with unpaired student t-test and some with One-way ANOVA. Graph has 3 groups of variables, on One-way ANOVA is more appropriated.
  2. Line 350: it is said that and increase in IL-B is detected at 6h p.i., but the significance is not represented in the graph. If it is not statistically significant, it cannot be called as an increase it has to be a tendency to increase or references like that.
  3. Line 353: “Significant increase in the production of IL-8”. The increase is not shown in the graph.
  4. Line 362: authors claim a non-significant increase of IL-8, but graphs say otherwise.
  5. Paragraph from line 430 to 434: Check that the statistics described in the text are correlating with the ** shown in the graphs.
  6. Figure 8B: It is not clear if rats explore more the novel object than the familiar one. This should be seen in order to assess if the test has worked.
  7. Paragraph from line 520 to 523: I do not see clear in the graph that the animals were treated; it just stands for Sham or UTI. Also just 2 groups, so if there are 2 treatments, there should be 3 groups at least. If there was treatment, how long before? Is maybe the figure referred in the text missing?
  8. Expression of intermediate early genes modified by UTI is recovered to normal levels in a time window of 24h, so can they really have an impact in long term neurogenesis?

Minor concerns:

  1. Supplementary Figure 1: PCR is not that clear. Some animals seem to be negative for MND-5 or some sort of degradation present.
  2. Names of the statistic test should be uniformed. Sometimes it stays unpaired student t-test and sometimes unpair two tailed student t-test or just student t-test.
  3. Line 414: Suppl. Figure 5A I think it is referring to Suppl. Fig. 6.
  4. Why supplementary figure 6 way before in the paper than suppl. Figure 4 and 5?
  5. Supplementary figure 5A: Scale used in the graph makes it difficult to compare differences between groups.
  6. Line 524: 34 days not 33
  7. Line 527: figure 8C not 8D (that makes me think that one figure is missing as said in point 7 of major concerns).
  8. Line 530: Figure 8D not 8E, figure 8E does not exist.

After including all these changes, the paper sould be reconsidered.

Reviewer 3 Report

Overall, this paper brings an interesting idea that there is a link between urinary tract infections and cognitive impairment in the central nervous system. The authors utilized various approaches to investigate the correlation between them. It would be beneficial for the audience who are interested in neurobiology, clinical research. But there are still some concerns and suggestions that might improve the paper to a better level.

Major changes:

  1. Overall, the paper was not well written, instead of explaining the results, the authors need to add the aims and what are results suggesting in the text (especially 3.2, 3.4, 3.5, 3.9, 3.11). For example: in result 3.3, the authors need to explain what’s the age of mice, what is “NeuN” staining for, what’s the aim of performing these experiments in the text?
  2. The study might be more concise if the authors can combine results 3.2 and 3.3.
  3. The flow might be better if the authors can move 3.4 and 3.5 to the place after the authors claim the behavior phenotypes found in UTI models in the result 3.12, it also would be more logic-based if the authors can combine the neurogenesis data with behavior data after drug treatments as final figures.

Minor changes:

  1. In figure 2, 3, 4, it would be more informative if the authors can provide the red and green channels separately, especially it was a bit difficult to see the colocalized signal in figure 3D
  2. The authors should label the green channel in the figure 2B
  3. The authors need to keep consistency when it comes to the figure number format (using “Fig” or “Figure”)
  4. On page 11, the authors mislabeled supplementary figure 5A in the text.
  5. The authors need to avoid double spaces in the text.
  6. Figures 8C, 8D are not consistent with what the authors claim in the text on Page 16
  7. Figure 8E is missing.

Round 2

Reviewer 1 Report

I meant instead of writing "Xyla" the author should write "Xylazine". Please delete the mechanism of action since it is irrelevant.

Please make all the figure legends standard by including the n fpr each condition and statistical analysis used.

Author Response

Comments and Suggestions for Authors

1. I meant instead of writing "Xyla" the author should write "Xylazine". Please delete the mechanism of action since it is irrelevant.

2. Please make all the figure legends standard by including the n fpr each condition and statistical analysis used.

Response to Reviewer:

We thank the reviewer for their comments and clarification.

For comment 1, we have replaced “Xyla” with “Xylazine” in the manuscript and deleted the mechanism of action.

In response to comment 2, we have now added the number of rats (n) per condition in figure legends and stated the statistical tests in the figure legends that they were missing from.

Reviewer 2 Report

The authors have properly replied all the major and minor concerns and gave good answers to all the questions.

With the new version, the manuscript has improved and thereby it could be already considered for publication,

Author Response

The authors have properly replied all the major and minor concerns and gave good answers to all the questions.

With the new version, the manuscript has improved and thereby it could be already considered for publication.

Response to Reviewer:

We thank the reviewer for their revision and their important previous comments.

Reviewer 3 Report

  1. Typos: in line 346: Figure21C &D
  2. Two “figure 2”, but no figure 1 in figure legends and the texts
  3. Figure format through the entire text is not consistent, for example, line 452: figure 7A and B which should be figure 7A&B, same as line 471

Author Response

  1. Typos: in line 346: Figure21C &D
  2. Two “figure 2”, but no figure 1 in figure legends and the texts
  3. Figure format through the entire text is not consistent, for example, line 452: figure 7A and B which should be figure 7A&B, same as line 471

Response to Reviewer:

We thank the reviewer once more for their important comments and revisions.

In response to comment 1, the typo in line 346 has been corrected to Figure 2 instead of 21.

In response to comment 2, the number of the first figure in the manuscript has been edited in the figure legend to become figure 1 instead of 2. As for the text referring to figure 1 it was correctly stated and thus no edits have been made to the text.

In response to comment 3, we have edited the manuscript accordingly to make the numbering consistent with using “&” instead of “and”.